# YouTube and the implementation and discontinuation of the oral contraceptive pill: A mixed-method content analysis

Jana Niemann[1,2]*, Lea Wicherski[3], Lisa Glaum[2], Liane Schenk[1], Getraud Stadler[4], Matthias Richter[2,5]

**1** Institute for Medical Sociology and Rehabilitation Science, Charité Universitätsmedizin Berlin, Berlin, Berlin, Germany, **2** Medical Faculty, Institute of Medical Sociology, Institute of Medical Sociology (IMS), Martin Luther University Halle Wittenberg, Interdisciplinary Centre for Health Sciences, Halle (Saale), Saxony-Anhalt, Germany, **3** Osnabrück University, School of Human Sciences, Osnabrück, Lower Saxony, Germany, **4** Institute for Gender Research in Medicine (GiM), Charité Universitätsmedizin Berlin, Berlin, Berlin, Germany, **5** Department of Sport and Health Sciences, Technical University of Munich, Munich, Bavaria, Germany

* jana.niemann@medizin.uni-halle.de

**Data Availability Statement:** Full data are available from the corresponding author or the Institute for Medical Sociology at the Martin-Luther-University

## Abstract

### Background

Women living in high-quality healthcare systems are more likely to use oral contraceptives at some point in their lives. Research findings have sparked controversial discussions about contraception in the scientific community and the media, potentially leading to higher rates of method discontinuation. Understanding the underlying motives for method discontinuation is crucial for reproductive health equity and future programming interventions. To address this question, this study aims to explore women's experiences of oral contraceptive use and discontinuation on YouTube.

### Methods

A concurrent explanatory mixed-methods design was used to conduct content analysis of German YouTube videos. The information from 175 videos of 158 individuals was extracted through quantitative descriptive content analysis. Twenty-one individuals were included in the qualitative content analysis.

### Findings

The body was a recurring theme in the pill biographies. Women described, for example, bodily sensations as reasons for taking and stopping the pill. They also described positive and negative side effects while taking the pill and after stopping. The most common side effects of taking the pill mentioned by YouTubers were mood swings (76/158), weight gain (45/158), headaches (33/158), and depressed mood (45/158). The symptoms after discontinuation reported most were facial skin impurities (108/158), decreased mood swings (47/158), hair loss (42/158), and weight loss (36/158). Overall, women overwhelmingly rated their discontinuation experience as positive (87/91).

Halle-Wittenberg (medizinische-soziologie@medizin.uni-halle.de) upon reasonable request.

**Funding:** This publication was supported by the Open Access Publication Fund of the Martin Luther University Halle-Wittenberg. The funders did not influence the data analysis and the results.

**Competing interests:** The authors have declared that no competing interests exist.

**Abbreviations:** IUD, Intrauterine device; OCP, Oral contraceptive pill; SNSs, Social networking sites; VTE, Venous thromboembolism.

## Conclusions

The study identified key symptoms of oral contraceptive initiation and discontinuation by portraying the experiences of female YouTubers, adding valuable insights to the understanding of method initiation and discontinuation. Further research is needed to explore women's personal experiences with method discontinuation beyond the YouTube platform.

## Introduction

The oral contraceptive pill (OCP) is a crucial contraceptive in high-quality healthcare systems in Northern and Western Europe [1–3]. Since its introduction 60 years ago, women have continued to report on the side effects of the pill [4]. Recent research has supported these claims, particularly regarding associations with side effects such as depression or venous thromboembolism (VTE) risk (e.g. [5–10]). However, most of these studies have reported associations, not causality. This has led to increasing criticism, especially by mainstream media [2,11–13].

### The pill and mainstream media–a problematic relationship?

**History of the relationship between the pill and mainstream media.** The dynamic between mainstream media and the OCP has had a long history: women's right to sexual self-determination and the OCP's side effects have been discussed in the media for more than half a century [14,15]. An important example is the "pill scare" phenomenon of 1995 [16–18], focusing on high-income countries. In this case, the dissemination and interpretation of scientific research regarding the risk of VTE associated with the OCP led to an increased number of discontinuations [16]. This was followed by individual contraceptive use and increased unintended pregnancies and abortions (e.g. in New Zealand [17], Britain [15], and Norway [18]). Media-critical discourse continued to emphasize the increased risk of VTE in the 2000s and early 2010s [15], while ignoring studies that found no increased risk [15]. This stance was also taken up in the OCP report [11] in Germany in 2015, which lacked scientific rigor [19]. The authors concluded there was a higher risk of thrombosis with the more modern 3rd and 4th generation OCPs. This was further disseminated by the media [20–22].

In 2016 and 2017, the media focused on the findings of increased risk of depression [23,24], based on two Danish studies [9,25]. These studies were, however, criticized in the scientific community due to methodological weaknesses [24]. Bitzer (2017), for example, criticized the lack of sensitivity analysis and questioned the biological plausibility [24]. Women shared their personal experiences with the OCP during this wave of OCP-skepticism under the #MyPillStory on social networking sites (SNSs) [26,27]. The VTE risk associated with the OCP was compared to that associated with the Vaxzervria vaccine (formerly COVID-19 Vaccine AstraZeneca) during the SARS-CoV-19 crisis. This is a difficult comparison because the VTE associated with OCPs is deep vein thrombosis and pulmonary embolism, whereas the VTE risk associated with the AstraZeneca vaccine is cerebral venous sinus thrombosis [13].

In summary, the media discussion of OCPs has underrepresented studies that do not show correlations with adverse side effects and have often failed to distinguish between absolute and relative risks [15,23]. The scientific analysis and evaluation of online contraceptive information has, therefore, become particularly important.

**Researching SNSs and hormonal contraception.** The relationship between media journalism and healthcare decision-making has increased significantly with the advent of the

Internet [14,28]. Greater access to online resources and SNSs has expanded the range of information available to women and increased their autonomy to choose contraception [13,14]. Hence, they have emerged as powerful tools for health communication.

This increase has also necessitated an exploration of the content, narratives, and quality of online information regarding (hormonal) contraception. Vieth et al. (2021) found that the internet was the most common source of contraceptive information among German women [29]. Studies, for instance, researched the experiences of birth control on YouTube [30–33], TikTok [34,35], and Reddit [36]. An analysis of German SNSs' contraception content on YouTube, Instagram, and TikTok found clear shortcomings in the quality and completeness of information on these sites [37]. Additionally, a study by Alves and colleagues [38] found that most of the websites mentioning the VTE risk associated with OCPs do not refer to information from accredited health agency sources. In terms of health communication science, so-called health laypeople (users of a contraceptive method) can disseminate information on the Internet [39]. Döring et al. (2023) showed that most German contraceptive content on YouTube, Instagram, and TikTok was uploaded by health laypeople, sharing personal contraception stories [37].

Informed choice regarding contraceptive methods is an important component of the Sexual and Reproductive Justice Framework [26,40]. Knowledge and information can contribute to achieving justice through increased reproductive agency and contraceptive choice [26]. However, the biased media attention and misinformation might lead to higher discontinuation rates among OCP users.

**OCP discontinuation.**   Contraceptive discontinuation can be defined as "*starting contraceptive use and then stopping for any reason while still at risk of an unintended pregnancy*" [41]. A German study on the knowledge and perceptions about OCPs among young women showed that 64.5% (n = 1480) of those who used OCPs at some point discontinued the method [29]. Simmons et al. examined the cessation of contraceptive methods as part of a person's *Contraceptive Journey* [42]. They defined key factors contributing to this decision: physiological factors, values, experiences, circumstances, and relationships (e.g. family, (sex) partners, friends, healthcare professionals). The discontinuation of OCPs deprives the body of an external source of artificial progesterone and estrogen, resulting in a change in the hormone levels in the body [43,44]. Current medical research on OCP discontinuation is limited to fertility restoration [45,46], fecundability [44], cycle characteristics [45,47], and reasons for discontinuation [29,42,48].

Contributing to the narrative of OCP discontinuation, Kissling explored posts of individual experiences on blogs and websites through a postfeminist lens [49]. Furthermore, Döring conducted a content analysis of German posts about the OCP on TikTok and YouTube [23]. She was interested in who the authors of the posts were, what the messages were about the OCP, and what the audience's reactions were. She found that most of the posts on YouTube came from health laypeople. She characterizes them as "pill-weary women" who give autobiographical accounts of taking and stopping the OCP. Viewers' reactions to these posts are mostly very positive, with many views and likes, very few dislikes, and lots of friendly comments. Adding to this, a recent study on birth control content on YouTube found that most women in 50 videos talked about their discontinuation experience with the OCP: the main outcomes after discontinuation were worsened acne (22%), improved mood (18%), cycle irregularities (14%), and increased energy (14%) [33].

However, the health consequences of discontinuing OCPs generally and on social media have not been adequately studied in the German context. There is also, as noted by Inoue et al., a lack of research on women's own experiences after stopping OCPs [48].

### Content analysis objectives and research questions

YouTube has established itself since 2005 as a way for individuals to share information and present themselves online. It is a video-sharing platform that is constantly changing [50]. Based on Döring's and Pfender and Devlin's analyses, YouTube videos uploaded by health lay-people, provide an opportunity to scientifically examine these personal experiences and determine the health consequences of initiating and discontinuing OCPs [33,39].

The overall objective of this content analysis was to investigate the personal physiological and psychological changes and lived experiences of German-speaking YouTubers after initiation and discontinuation of OCP treatment. We, therefore, aim to examine

- the reasons for starting and discontinuing the use of the OCP,

- to document any side effects experienced during and after use, and

- how the women describe their history with the OCP.

## Methods

### Design

The study follows the Open Science movement, i.e. the pre-registration, and all data are stored on the Open Science Foundation server (https://osf.io/fekdh/). It was designed as an (almost) simultaneous exploratory mixed-methods content analysis [51]. The quantitative component was primarily used to present overall video characteristics and estimates of video content data. The qualitative component explored YouTubers' perceptions and beliefs as an in-depth analysis to complement the quantitative research. The qualitative and quantitative research strands were initially analyzed separately and then brought together for the interpretation and presentation of the results.

### Sample

YouTube was initially searched from July 20 to 23, 2021. An updated search was performed on May 5, 2023. The internet browser and cookies history were cleared before the search to avoid biases from our laptop. The sample was examined by using common searches for videos of German-speaking women who explained their personal experiences with the discontinuation of the OCP. YouTube was searched using the autofill feature in the search bar (which uses an algorithm influenced by users' popular searches to automatically fill search queries with root words) [52]. The following translated search terms were used: "stop taking the pill," "stop taking the pill experience," "discontinuation of the pill," and "discontinuation of the pill experience" (the exact German terms are provided in [S1 Table]). This search was performed following previous YouTube analyses (e.g. [50,52,53]). The videos were sorted by the relevance filter (= year). All videos were included for each search term. They were then checked for duplicates.

Only videos that met our inclusion criteria, as shown in Table 1, were included.

If a YouTuber repeatedly uploaded more than one video, two to three videos were included: The first uploaded video (most likely to contain relevant information about the reasons for discontinuation or duration of OCP use) and the second and/or last uploaded video (probably the most recent version regarding immediate health consequences). If a YouTube video was uploaded with two women talking about their experiences, both individuals were included in the study. Different personal identifiers (see ethics) were assigned.

**Table 1. Eligibility criteria.**

| Inclusion criteria | Exclusion criteria |
|---|---|
| 1) YouTube videos with a major focus on the discontinuation of the OCP and its consequences.<br>2) YouTubers talk about their personal experiences with stopping the OCP.<br>3) The author herself must be the person talking about her experiences in the videos.<br>4) Video data content<br>5) Videos with a minimum length of 5 minutes.<br>6) German YouTube videos. | 1) YouTube videos where there is no major focus on the discontinuation of the OCP (video formats such as Vlogs).<br>2) YouTubers that report on non-personal content that is (not) related to stopping the OCP.<br>3) A third party talking about the discontinuation of the OCP.<br>4) Other media formats, such as podcasts.<br>5) Videos shorter than 5 minutes.<br>6) Non-German YouTube videos.<br>7) SHORTS, a short video format on YouTube. |

Fig 1 shows the flowchart of the search. A total of 1344 videos were initially collected by JN. After removing all duplicates, 591 videos were reviewed by JN and LW. After removing an additional 416 videos, a final number of 175 videos from 158 YouTubers were included in the sample for quantitative content analysis. All individuals could be identified as female.

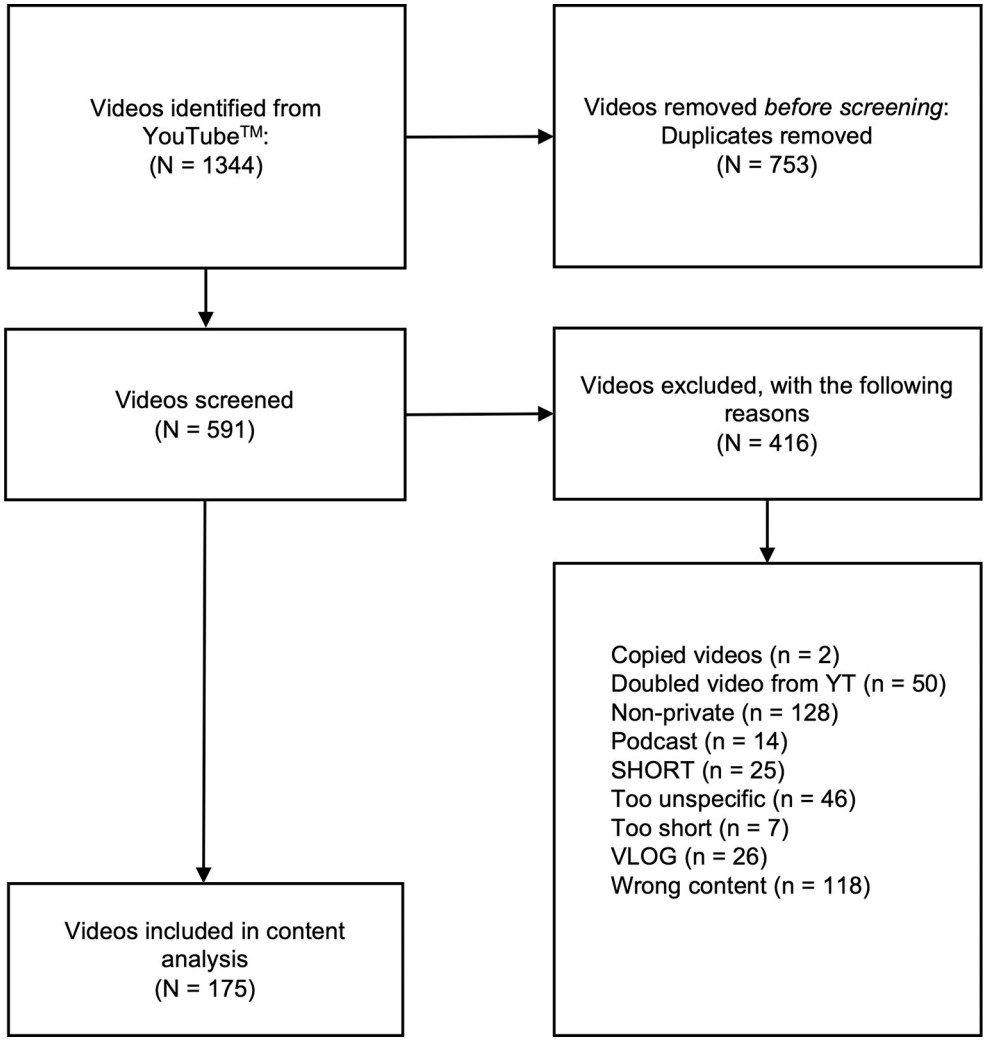

**Fig 1. Flow diagram of the video selection process.**

Hereinafter, 18 videos were included as an initial sample for the qualitative content analysis. These were selected by JN and LW after reviewing all 175 videos, based on stratification factors, to represent the variety of experiences of the study sample:

- reasons for initiation and discontinuation,

- length of initiation,

- age at initiation and discontinuation, and

- experienced side effects during intake and after discontinuation.

After JN and LW coded the initial sample, three additional videos were included to reach the point of data saturation [54]. This resulted in a total qualitative sample of 21 videos.

## Procedure and analysis of the content analysis

The uploaded videos (audio and text presentation) and the person(s) who reported on their personal experiences with stopping the OCP on the online platform YouTube were the units of analysis for the media content.

**Procedure and analysis of the quantitative content analysis.** A predefined, theoretically driven codebook (S1 File) was developed. The quantitative content analysis consisted of coding content data related to the video (e.g. V001) and the person(s) reporting their experience (e.g. P001):

1. Typical YouTube **video characteristics**, including posting date, video length, number of views, likes, and comments.

2. **General content data regarding the OCP persistence and discontinuation**, such as the time of video recording after discontinuing the OCP (weeks), length of OCP persistence (years), age at initiation and OCP discontinuation, second discontinuation, reasons for initiation and discontinuation (0 = not mentioned, 1 = mentioned), and current contraceptive method (0 = not mentioned, 1 = mentioned).

3. **Content data regarding the OCP implementation**, including the mention (0 = not mentioned, 1 = mentioned) of improvements (e.g. facial skin) and side effects (e.g. migraine).

4. **Content data regarding the OCP discontinuation**, including the mention (0 = not mentioned, 1 = mentioned) of the physiological and psychological changes (e.g. mood swings).

The translated codebook is available as (S1 File). JN and LG independently assessed the intercoder reliability of the codebooks based on 17 randomly selected videos from the sample. The Krippen-Dorff's Alpha reliability coefficient was calculated for all content variables in the codebook. The following average reliability value was obtained for the codebook: 0.813, indicating good measurement accuracy [55].

Data was collected by downloading the videos and the text presentation. These were transferred directly into MAXQDA 2020, where the material was coded by JN. The data were then transferred to an Excel spreadsheet. Quantification was performed by JN using descriptive data analysis (frequencies and percentages) in STATA 14.2.

**Procedure and analysis of the qualitative content analysis.** This analysis consisted of coding the content data related to the person sharing their experiences (e.g. P001) in the uploaded video. After selection, the videos of YouTubers were transcribed verbatim. Their content was then analyzed by JN and LW according to Mayring's qualitative content analysis [56]. A category system was developed beforehand, according to which, the content-related aspects of the material were examined. The category system is available as (S2 Table).

The categories were developed based on the current state of research and by considering the codebook of the quantitative research strand. Further subcategories were created based on the research questions and inductively from the material. Coding rules were subsequently defined, and anchor examples were added. The videos were transcribed and coded by LW and JN in MAXQDA 2020.

*Reflexibility.* The two main researchers involved in the qualitative research (JN and LW) have taken and stopped the OCP themselves. Although we were not required to participate in interviews, our lived experiences were part of the knowledge production in this study. Both researchers talked about the process of coding and creating categories throughout the data analysis.

## Ethics

The ethical approval for this study has been granted by the Ethical Review Committee of the Medical Faculty at Martin Luther University Halle-Wittenberg. The German Society for Online Research [57] states that the use of publicly available data for scientific research is ethically appropriate even without informed consent. Before a person uploads a video on YouTube, they have to consent that their data can be reused through third parties [58]. Therefore, a certain level of awareness can be assumed [59]. We are aware that this requires a certain level of maturity. Therefore, we decided to only include adult YouTubers who are aware of their publicity (e.g. through follower numbers) and the publicly available works.

All videos and individuals were pseudonymized at the beginning of the data collection to minimize the risk of identification. This makes it impossible or very difficult to identify them in the research process: All videos were assigned a video identification number (e.g. V001). All people have a personal identification number (e.g. P001). The personal data (e.g. title, YouTube channel name) were separated from the factual data (e.g. time of posting, views) and stored on different media (USB key).

The collection and analysis method complied with the terms and conditions for the source of the data.

## Results

The 175 Videos analyzed in this study were posted between 2014 and 2023. The mean video length was 14.8 minutes. The detailed general characteristics are available in the (S3 Table).

### General characteristics of the OCP persistence

The mean duration of OCP use was 8.2 years. Women started using OCPs at a mean age of 14.8 years. The mean age of discontinuation was 23.1 years. See Table 2 for details.

**Table 2. General characteristics of OCP persistence.**

| Characteristics | Min | Max | Median | Mean | Standard deviation |
|---|---|---|---|---|---|
| **Time since discontinuation (weeks)** (N = 120) | 0.0 | 108.0 | 12.0 | 15.9 | 19.4 |
| **Persistence** | | | | | |
| Length (years) (N = 119) | 1.0 | 27.0 | 7.0 | 8.2 | 4.3 |
| Age initiation (N = 119) | 12.0 | 23.0 | 15.0 | 14.8 | 1.4 |
| Age discontinuation (N = 95) | 17.00 | 37.00 | 22.7 | 23.1 | 3.5 |

### The OCP implementation and discontinuation

The following results show that embodiment and effects on the body play a central role in OCP use and discontinuation. This recurs in the detailed descriptions of the psychological and physical aspects of each phase of the OCP autobiographies uploaded.

**Characteristics of the OCP implementation.** Table 3 details all the information on the OCP implementation. A total of 57 of 158 YouTubers changed their type of OCP during the implementation period. Aspects of this are that switching may have occurred because of side effects of the OCP (P018, P149), side effects that occurred after switching (P010, P100), or side effects persisted and worsened after switching (P149).

The reasons for taking the OCP were varied and included facial skin impurities (35/158), contraception (30/158), and/or painful menstruation (27/158). Many felt that the OCP was the only option available to them and were unaware of alternative contraceptive methods. The women expressed the normality of the OCP as a solution for treating facial skin impurities (P002) and the gynecologist's recommendation for easing menstrual cramps (P010). According to the average age at initiation, it was stated that pregnancy was not an option due to the young age and that the OCP was considered a safe method of contraception (P006, P018, P149).

**Positive side effects during implementation.** Overall, 45 of the 158 women experienced improvements while using OCPs. The improvements were consistent with the main reasons for initiation: facial skin impurities (25/158), painful bleeding (19/158), and decreased bleeding (12/158). The OCP appears to be particularly convenient as it combines the treatment of medical problems, such as painful bleeding, with the need for contraception (P007, P010).

**Negative side effects during implementation.** Physiological side effects were reported by 101 of the YouTubers in the study. The most common side effects were weight gain (45/158), headaches (33/158), and water retention (30/158). Headaches seemed to be very distressing and interfered with women's lives (e.g. P001, P007, P100). A total of 103 women reported psychological side effects. The most common psychological side effects were mood swings (76/158), depressed mood (45/158), and deterioration of libido (31/158). Mood swings were characterized as being moody, irritable, unrelaxed, aggressive, or angry (e.g. P010, P032, P054). Women also reported feeling that something was missing or feeling emotionally numb while on the OCP (e.g. P032, P085, P148).

### Characteristics of the OCP treatment discontinuation

As described in Table 4, the main reasons given by YouTubers for discontinuing the use of OCPs were the side effects experienced (74/158), increased awareness of the topic (21/158), and the desire to stop taking hormones (34/158). Qualitatively, side effects that led to discontinuation could include a tipping point where a user could no longer tolerate the side effects. In particular, the severe impact of headaches while taking the OCP was reported as a side effect leading to discontinuation (P001, P007, P018). A total of 27 of all YouTubers in the study reported discontinuing the OCP for the second time. The reasons for resuming the OCP were changes (e.g. facial blemishes) after the first discontinuation. However, as one YouTuber explained in P018, the side effects of the OCP were even worse, which then led to the final discontinuation of the method. Ultimately, former users of the OCP had a variety of reasons for stopping, but all prioritized their overall health and well-being.

**Improvements after discontinuation.** Out of 158 YouTubers, 54 mentioned physiological improvements and 82 mentioned psychological improvements from discontinuation. The most common improvements mentioned were weight loss (37/158), the disappearance of headaches (23/158), decreased fatigue (13/158), decreased mood swings (47/158), increased libido (40/158), and decreased depressive mood (19/158). Women reported that

**Table 3. OCP implementation.**

| Generation | Frequency % (n/N) | Example (Qualitative) |
|---|---|---|
| **Switch between OCPs** | 36.1 (57/158) | "He prescribed me another pill and from then on I only had problems with the pill. All the side effects that I had before were much worse and I only really noticed them once." *(P010)* |
| **Reasons for initiation**\* | | |
| Facial skin impurities | 22.2 (35/158) | "[. . .] I really took it at that time because I had bad facial skin and at that time it was just like, if you have bad facial skin, just take the pill. And nobody thought too much about it, it was just normal and of course parents were like, okay, then take the pill before you get pregnant or something. And my facial skin actually got better and I was happy [. . .]." *(P002)* |
| Contraception | 19.0 (30/158) | "After my first menstruation, I started taking the pill, and at that time also for the purpose of contraception. Before that, I had already tried using a condom, but to be honest, I didn't really feel safe with it myself." *(P149)* |
| Painful menstruation | 17.1 (27/158) | "And for me, it wasn't about contraception at all, but I had very bad stomach pains every time I had my period, so I missed about three days a month. And like so many others, my gynecologist recommended the pill." *(P010)* |
| Gynecologist | 5.1 (8/158) | / |
| Other | 17.1 (27/158) | "A lot of people may know that I had an abortion when I was 16 and I didn't say okay; I never want to have to go through what I went through there again. That's why I then started taking the pill." *(P148)* |
| **Improvements**\* | 28.5 (45/158) | / |
| Facial skin impurities | 15.8 (25/158) | "[. . .] I got better facial skin quickly, although I have to say that when I went to the gynecologist, I also said that I would like to get a pill that would really improve my facial skin. So, I mean, I got a pill against acne and, therefore, my facial skin quickly improved accordingly and that was of course totally great." (P100) |
| Painful bleeding | 12.0 (19/158) | "I noticed right after the first month that my period was a lot less and it hurt a lot less and I was like oh my God (.) thank you that they invented something like this because it was really bad for me I had really bad pain." *(P001)* |
| Decreased bleeding | 7.6 (12/158) | "Of course, the pill also had some positive effects. I had less heavy periods." *(P010)* |
| Other | 3.2 (5/158) | / |
| **Physiological problems**\* | 63.9 (101/158) | / |
| Weight gain | 28.5 (45/158) | "I had gained weight once because of it. Like I said, it didn't bother me that much. But that was already about eight kilos that I had gained. Just steady, steady, so there was no stopping it." *(P018)* |
| Water retention | 19.0 (30/158) | "So a week before my period, I had so much water retention [. . .]." *(P079)* |
| Weight gain breast | 8.9 (14/158) | / |
| Headache | 20.9 (33/158) | "But at some point when I was 17, I noticed that I had extremely bad headaches all the time. Really, it was so bad for me. I used to just think I stayed home for two days because I had such stomach pains and at that point at 17 I stayed home because I had such bad headaches that I thought I had a migraine, what's wrong with me?" *(P007)* |
| Migraine | 9.5 (15/158) | "So, in general, I have to say that with the pill, I had constant headaches, I really suffered from migraine attacks." *(P140)* |
| Exhaustion | 9.5 (15/158) | "[. . .] when I was on the pill, I was super tired, super floppy, and really fell asleep at 9:30 in the morning because I was just so exhausted." *(P048)* |
| Spotting | 5.1 (8/158) | / |
| Pain legs | 5.1 (8/158) | "I have always had severe cramps in both legs. When I sat for a long time, it was like an endless cramp in my leg. It felt like dogs were biting me. So yes, but of course I thought it was normal." *(P140)* |
| Other | 43.7 (69/158) | "I felt sick a lot. I had this nausea a lot, and I don't know, it was all kind of weird, and I thought about it a little bit and I thought, okay, what could it be?" *(P007)* |
| **Psychological problems**\* | 65.2 (103/158) | / |
| Deterioration of libido | 19.6 (31/158) | "Loss of libido, for example, is so extreme with me that I actually have no desire. So, I take this contraceptive and I have no desire for sex. So, you think, okay, what am I taking it for? Because if it doesn't do what it's supposed to do, I don't know, it's questionable." *(P119)* |
| Depressive moodiness | 28.5 (45/158) | "And that was what was missing. I often felt in very, very positive situations, when you should feel very, very happy, that something was missing. So, I knew rationally, okay, I should be very, very happy now, I should be totally happy now, but inside I just didn't feel it and it was totally weird. But then a lot of times I just blamed it on the fact that I'm just getting older and maybe you just get a little bit of a number when you get older." *(P149)*<br>"I don't know if the depression I got when I was 19 was triggered by the pill. But there is a good chance that I am one of the 23 % of women who suffer from depression triggered by the pill." *(P010)* |
| Mood swings | 48.1 (76/158) | "I was super quick to get irritated; I was super quick to get bitchy when I didn't like something. So, I'm not really like that and I didn't want to be. But somehow, I did. The pill just made me super irritable and super moody before my period." *(P079)* |

*(Continued)*

**Table 3.** (Continued)

| Generation | Frequency % (n/N) | Example (Qualitative) |
|---|---|---|
| Other | 11.4 (18/158) | "But then when I was 18, 19, I think, it started so slowly that I developed anxiety and I started having panic attacks and my body felt kind of weird." *(P044)* |

*Overlap possible.

discontinuation improved their psychological mood: They felt more relaxed and had a more positive attitude (P001, P002, P036, P048, P054). They also reported a return or increase in their libido (P002, P100, P140, P149). Many expressed that they did not expect how intense this return would be (P010). They reported feeling more in tune with their bodies and experiencing a greater sense of euphoria after weaning off the OCP.

**Negative changes after discontinuation.** In general, 123 of all women in the study experienced negative physiological and 32 experienced psychological changes after discontinuation. An increase in facial skin impurities was the negative physiological change most reported after discontinuation (108/158). This was followed by hair loss (42/158), painful menstruation (36/158), and menstrual cycle irregularities (31/158). Negative psychological changes included mood swings (16/158) or the onset of premenstrual syndrome (10/158). The YouTubers claimed that their facial skin impurities were never as bad as they were after stopping the OCP. They mostly described their facial blemishes as "normal" and not as severe as in people with "real" acne. Conversely, they also described the onset of facial skin deterioration as a major challenge and burden. Due to the strong identification with their facial skin and external appearance, this had become a problem. Thus, good make-up and the thought that others also have facial skin imperfections should help to conceal and accept one's facial skin, which is seen as a flaw (P002, P018, P100). A minority of women stated that discontinuing the product caused a worsening of their psychological state (P085).

**Evaluation of contraceptive discontinuation.** Overall, 87 of 91 YouTubers rated their experience of discontinuation as positive. The women reflected on the strong influence of the OCP on their bodies and health (P002, P010). They described how happy they were to have freed their bodies from artificial hormones (P079), and that they were more in tune with their bodies and more aware of themselves (P010, P044, P100, P149). Words that appeared frequently in the conclusions were "liberated" and "freer." Women felt freer after weaning off the OCP, for example, even though they said it was difficult to put this into words (P032, P036, P054).

**Current contraceptive method.** The most common contraceptive currently used by YouTubers is the condom (26/84). This is followed by the copper intrauterine device (IUD), chain, or ball (24/84) and the natural family planning (NFP) method (14/84).

## Critical views on the OCP treatment

Looking back on their time on the OCP, women also expressed critical views. It was emphasized that the first sexual experiences in life were made under the influence of the pill (P010, P044). As one YouTuber explained:

> "One of the things I think is very fatal about very young girls being prescribed the pill is that many start taking it before they even have a sex life. That's exactly what happened to me. That is, every sexual experience I had was under the influence of the pill, so I didn't even know what sex and my sexuality would be like without the pill. So, for years I thought a lot of things were normal." (P010)

**Table 4. OCP discontinuation.**

| Generation | Frequency % (n/N) | Example (qualitative) |
|---|---|---|
| **Reasons for discontinuation*** | | |
| Experienced side effects | 46.8 (74/158) | "For one thing, I was really super tired and totally exhausted a lot of the time, even though I was eating healthily. [. . .] I had these headaches, and I don't know, for me it was just like there were so many things and so many issues. I always had this tight feeling in my chest and there were so many issues and thoughts in my head that I just really said, you know what, from one day to the next I'm going to take the pill all the way through this month, then I'm going to wait until I get my period and then I'm not going to take the pill anymore." *(P007)* |
| Increased awareness | 13.3 (21/158) | "The reason why I stopped was because I saw it with so many other people. So it was just kind of such a hype in the blogger world and in the YouTuber world that you stop taking the pill because yeah, because they didn't want to take hormones anymore. And then I heard that a lot of people were super happy with it and that a lot of people felt liberated and just had a much better mood and then I thought to myself, this is exactly what I want." *(P036)* |
| Fear of side effects & long-term consequences | 8.9 (14/158) | "Let's put it this way, and especially if I were in my own facial skin, if I were affected by it myself, I wouldn't want it. I don't want thrombosis; I don't want hypothyroidism." *(P083)* |
| Exchange with friend | 7.0 (11/158) | "I didn't come up with it myself, but my boyfriend at the time sent me an article and said, 'Hey, somehow it turns out that a lot of pills increase the risk of thrombosis quite a bit. Check if your pill is one of them.'" *(P010)* |
| Desire to stop taking hormones | 21.5 (34/158) | "[. . .] when I was no longer in a relationship, I decided I just wanted to stop taking the pill and see how my body felt without the hormones." *(P149)* |
| Own health | 5.0 (8/158) | "I did it just for me, so just for me and my health and my body and so on." (P006) |
| Other | 26.6 (42/158) | "It's not a great feeling. You don't really feel like a woman. It is. It's you. That's a really weird way to put it. I don't think you can say that you don't feel like a woman, for God's sake. But something is missing. There's something missing. And you can't put your finger on it because you don't know exactly what it is anymore." *(P083)* |
| **Second discontinuation** | 17.1 (27/158) | "At some point, however, I started having problems that were very noticeable. I'll get into that in a minute, and then I just stopped taking it. A little later I started with another pill, which was also a much weaker dose. But I had severe problems again, extreme headaches, and then I stopped taking it. And that was it for me and this method of contraception. I never took the pill again." *(P018)* |
| **Improvements*** | | |
| *Physiological* | 34.2 (54/158) | / |
| Weight loss | 23.4 (37/158) | "And I could definitely see how I lost those eight pounds on my stomach, thighs, and face in those eight weeks." *(P018)* |
| Decreased fatigue | 8.2 (13/158) | "Since I stopped taking the pill. It's only been about half a year that I'm really up super late, I'm doing really well in the evenings, and I don't feel as drained as I used to." *(P048)* |
| Disappearance of headache | 14.6 (23/158) | "And then what I also noticed enormously after I stopped taking the pill was that my headaches were gone. [. . .] I always had headaches and I always took pills because I couldn't stand it. When I stopped taking the pill, I didn't have headaches anymore. I don't have headaches anymore [. . .]." *(P002)* |
| Disappearance of migraine | 6.3 (10/158) | "Because when I stopped, it was all gone. I never had migraines again, and I had them really, really badly, so I just couldn't leave the house all day on the first day of my period, I had to lie in bed in my dark room." *(P036)* |
| Other | 6.3 (10/158) | "So, first of all, it was definitely relatively quick after I stopped the pill that any side effects I had were gone. So the eye twitching was gone." *(P018)* |
| *Psychological* | 51.9 (82/158) | |
| Increased libido | 25.3 (40/158) | "About a month or two after stopping the pill, my libido finally came back [. . .] I didn't expect it to be so extreme. It was like one day to the next. So from no desire at all to such a strong desire [raises arm above head] and I didn't even know I could have such a libido. Without going into too much detail now, but I had an almost constant desire for sex." *(P010)* |
| Decreased mood swings | 29.7 (47/158) | "I also had very, very blatant mood swings in the early days. [. . .]. In the meantime it has really gone away and I also notice [. . .] everything is cool. And I'm really relaxed [. . .] so it's no longer this, this 180 feeling in me, it's so really so okay." *(P001)* |
| Decreased depressive moodiness | 12.0 (19/158) | "From one day to the next, I was no longer sad, or simply no longer in a depressed mood. [. . .]. I was happy again. I could feel joy again." *(P036)* |
| Other | 2.5 (4/158) | / |
| **Problems** | | |
| *Physiological* | 77.9 (123/158) | |
| Hair loss | 26.6 (42/158) | "I had real clumps of hair in my hand after washing my hair, and there was hair everywhere. Until I had real bald spots on my head [. . .]." *(P036)*. |

*(Continued)*

**Table 4.** (*Continued*)

| Generation | Frequency % (n/N) | Example (qualitative) |
|---|---|---|
| Greasy hair | 15.2 (24/158) | "[. . .] my hair got greasy faster." *(P010)* |
| Facial skin impurities | 68.4 (108/158) | "Sometimes I have to fight so hard with my facial skin. So even when I wear makeup now, you don't see it so much. But some days, for example, I only have three pimples and I go to sleep and wake up the next day and suddenly I have new pimples. It's just not a joke, it's so blatant sometimes. [. . .'] I think it's a bit of a shame that I identify so much with my facial skin. So I always try to really let it go, because I think to myself: 'This is just a facial skin and no idea, my facial skin is not supposed to be perfect.' So, I mean, also, it's such a difficult subject, I think. Because, on the one hand, you see people with perfect facial skin all the time and you and maybe they use contraceptives. But I always try to isolate myself a little bit from that [. . .]." *(P100)* |
| Oily facial skin | 8.9 (14/158) | "I got oily facial skin, I got really shiny. I didn't get any pimples, thank God." *(P001)* |
| Hormonal acne | 5.7 (9/158) | / |
| First period | | |
| 1st month | 22.2 (35/158) | / |
| 2nd–5th month | 6.3 (10/158) | / |
| 6th–12th months | 3.2 (5/158) | / |
| Cycle irregularities | 19.6 (31/158) | "At first nothing happened. Really nothing. I didn't feel any different. I had no bleeding, nothing. I waited for two or three months and then at some point I went to my gynecologist and asked what was going on. My gynecologist was actually very calm and cool." *(P010)* |
| Strong menstruation | 13.9 (22/158) | "Of course, the period is heavier, but it is within a healthy range, I would say. So not that it would be extremely negative, but everything is still in the green zone, I would say." *(P048)* |
| Painful menstruation | 22.8 (36/158) | "[. . .] I had unbelievable pain again when I had my period." *(P085)* |
| Migraine | 10.3 (15/158) | "The first week was really weird. I had such bad migraines, my head was completely full. I got up every day with a migraine." *(P140)* |
| Hair growth | 5.7 (9/158) | "I've also noticed that my body hair has increased in places where I didn't really have any before. I've got such light beard hair, and also my hair on my legs, etc. has kind of grown faster." *(P010)* |
| Other | 21.5 (34/158) | "I got cysts in my breasts, not one or two, but sometimes three in one breast, but they were not bad, just deposits in my body." *(P119)* |
| *Psychological* | 20.3 (32/158) | / |
| Mood swings | 10.1 (16/158) | "I felt that all those side effects that most people have from the pill, like mood swings and even mild depression, came to me the moment I stopped taking it. Well, not right away, of course, but after a few weeks or months when I stopped taking the pill. I didn't feel like doing anything. I just felt really bad. I felt such a complete lack of drive and I didn't want to do anything. I didn't want to see anybody." *(P085)* |
| Premenstrual syndrome | 6.3 (10/158) | "Now that I'm off the pill, I feel but the last times I had my period always mood swings, but so negative somehow stop that sometimes I don't want the Stefan comes too close to me. Everything is often too much for me during my period. [And after my period I am the most relaxed and happy person again. So yeah, I find it pretty intense and I just don't know that about myself from before." *(P048)* |
| Other | 12.7 (20/158) | / |
| **Conclusion discontinuation** | | |
| Negative | 3.3 (3/91) | "I finally went back on the pill after about four months because I was feeling so bad physically and psychologically that I figured I'll just try it and if it doesn't work then I'll have to change something else anyway. The bottom line is that now, two, three months later, I'm fine. So, I don't know, maybe it's just a placebo effect. Maybe it's something completely different or maybe my body just can't do without hormones anymore and has gotten so used to it that it somehow doesn't work without them. Which is not healthy either, I know. So this is definitely not a permanent solution. [. . .] But at least now it is my solution until I find something else. All my problems are definitely gone. I feel much better physically and mentally." *(P085)* |
| Indifferent | 1.1 (1/91) | / |
| Positive | 95.6 (87/91) | "So the conclusion I can draw from all of this is [. . .] I don't regret it at all because my body is free of artificial hormones, I don't have headaches anymore, I'm calmer, I'm more relaxed, I'm much fitter, I feel better, I'm definitely healthier. I definitely know that it's much healthier for my body not to be pumped full of hormones every time [. . .]." *(P002)* |
| **Current contraceptive method** | | |
| Condom | 31.0 (26/84) | "[. . .] that's why I don't have any other contraceptive method like a condom now. I actually like it quite a lot. With a condom, you can always check to see if it leaked or not. And the morning-after pill would still be a hormonal option, but okay. I have to live with it now if I don't want to get pregnant." *(P083)* |

(*Continued*)

**Table 4.** (Continued)

| Generation | Frequency % (n/ N) | Example (qualitative) |
|---|---|---|
| Copper IUD/Chain/Ball | 28.6 (24/84) | "The most important thing for me personally in a contraceptive method, besides the fact that it should be hormone-free, is safety, and I came across the T-coil and the copper IUD and decided to use them." (P018) |
| NFP | 16.7 (14/84) | "[. . .] you can also use NFP for contraception. And because of the security that I had gained, I knew that the method really worked. Because I was honestly a little skeptical at first whether it would work so well with the body tracking, but then I was able to determine it, as I said, totally well, and that was honestly a pretty cool feeling. Yes, I talked to my partner about using it for contraception. I explained the method to him because it was very important to me that he knows what I'm doing and that he decides that it's okay for him that we use the knowledge for contraception. And since then we have been using NFP for safe, hormone-free contraception, which means that we have sex without a condom on the non-fertile days and with a condom on the fertile days." (P149) |
| Other natural contraceptive method | 9.5 (8/84) | "We actually just do it by calendar. That means I look at when I ovulate. I look at when I could get pregnant, when it is most likely, and I abstain from sexual intercourse completely." (P140) |
| Other | 14.3 (12/84) | "I am currently back on the pill. [. . .] Because I met my boyfriend back then [. . .]." (P119) |

*Overlap possible.

The pill was also seen as a moneymaker for the medical/pharmaceutical market (P032) and as a beauty product (P119, P149). The method of prescribing hormones to young girls at the beginning of puberty was questioned. They YouTubers also criticized the fact that the OCP was/is prescribed as a solution to non-contraceptive problems, such as facial blemishes or cycle irregularities. As this woman explained:

"And then I really wonder if it's necessary to try to influence that with drugs and hormones. Or,

maybe you could approach it differently or go in the direction of facial skin care and see a dermatologist to see if that might help [. . .]. Instead of just taking these hormones, especially at a young age, to get a better picture, I find it a little bit critical, to be honest, because I always think to myself, I don't know, it's coming from something else in your body. And you're just not treating the cause, you're just treating the problem at that moment." (P119)

The general cost of contraception as a female burden was also mentioned. P130 described that:

"Of course, it costs money, and you shouldn't forget that either. It's also really absurd that we women have to pay for it and the men don't. Not because of men and women, but I think the health insurance should pay for it. I can't even think about paying that much a month. And I never go to the doctor. Never. I never have anything. I don't even know what for."

What is interesting here is that they see the financial burden as something that should be negotiated and shared between men and women. She goes one step further and demands that health insurance should cover contraception. Women described different aspects of stopping the OCP in several videos.

## Recommendations regarding the OCP

The first recommendation was that women should not stop taking the OCP without information and indiscriminately (P018, P050). Women should make up their minds, seek medical advice (P002, P007), and not stop taking the OCP overnight, especially if it is being used for

contraception. It was also mentioned that both starting and stopping the OCP is a very individual matter (P018, P032).

The second recommendation emphasized that discontinuation is an individual decision. The YouTubers explained that the choice of contraception is a private matter (P006). You should not be talked into it (P018, P054) and make your own informed decision:

"It doesn't have to be for everyone and I would say to you don't let anybody talk you into something if you feel comfortable with it and it doesn't have to be the pill, it actually applies to all situations in life. If you feel comfortable with something and you think it works for you, then stick with it. Don't let anybody talk you into something that you don't feel is right for you." (P085)

Finally, some YouTubers also recommended stopping the OCP (P140, P010). They were extremely positive about their experience of stopping, describing how it is better to do without:

"That's why I can advise every woman and every girl to think about whether you take the pill, whether you start taking it or whether you continue taking it. Do some research. Pay attention to your body, pay attention to possible side effects, and think twice. So if you are thinking about going off the pill, I wish you perseverance and I promise you it will have a positive impact on your life and you will not regret it." (P010)

## Discussion

This analysis demonstrates that bodily experiences and the body itself are significant components in OCP autobiographies, from initiation to the period after discontinuation. The primary reasons for initiation are to address facial skin impurities, use as a contraceptive method, and to alleviate painful menstrual cramps. Common side effects experienced during the use of OCPs include mood swings, weight gain, headaches, depressed mood, decreased libido, water retention, and migraines. Women have described these effects as burdensome, despite the OCPs serving both medical and contraceptive purposes. Women predominantly discontinue OCPs due to side effects, a desire to cease hormone intake, and increased awareness of related issues. After discontinuation, individuals commonly reported deterioration in facial skin impurities and hair. But noted weight loss, reduced mood swings, increased libido, and a generally positive experience of feeling more connected to their bodies and freer. To the best of the authors' knowledge, this is one of the few studies on the personal views and experiences of women who have discontinued the OCP. The combination of quantitative description and qualitative analysis in interpreting the results was not only helpful in measuring experiences with the OCP, they showed how these experiences are perceived and lived. The results of this study are consistent with scientific research on the positive and negative side effects [60–62] of OCPs, the reasons for discontinuation [29,48,63], and embodiment [64,65].

Puberty is the time when most girls in quality healthcare systems visit a gynecologist for the first time. This may be for specific health problems, such as menstrual cramps, acne, and cycle irregularities, or contraceptive advice [66,67]. In our sample, most women initiated the OCP during puberty, an age when physical and hormonal development from girl to woman is not yet complete. The OCP, therefore, acts as a drug in an organism that is not yet fully developed [68]. In this context, health professionals must provide information and advice on the costs and benefits of contraception.

The side effects reported by the YouTubers during OCP treatment are generally consistent with the associated side effects reported in other studies [43,61,69] and listed on the OCP's package inserts [70]. However, the high number of side effects experienced by YouTubers was striking. This may be because YouTubers may be more likely to report discontinuing the pill if they have experienced side effects while either taking or discontinuing the OCP. Thus, there may be a publication bias in the YouTube videos and the results should be interpreted with caution.

The strata of this analysis are characterized by their young age at quitting. This may reflect the medium of YouTube as a YouTube consumer, which is specific to the sample of this study. However, it could also reflect the fact that women stop using the OCP at a relatively young age, particularly compared to the average age of women at first birth in the European Union (29.4 years) and Germany (31.2 years) in 2019 [71].

The main reasons for discontinuing OCP use reported were side effects and the desire to stop taking hormones. This is consistent with the quantitative literature, where side effects [29,48,63] or concerns about long-term effects [29] are predominantly cited as the main reasons for discontinuation. Additionally, Pfender and Devlin's analysis (2023) reported that 22 out of 50 YouTubers discontinued hormonal contraception in favor of "being more natural" [33]. A systematic review of medical and epidemiologic studies in 2015 [48], which examined OCP discontinuation, found that pregnancy was among the top two reasons for discontinuation. This discrepancy could be explained by the young age at discontinuation of the sample included in this study. However, it may be some of the other reasons for discontinuation mentioned above that prompted the women to create a YouTube video.

Scientific studies show that women especially are under social pressure when it comes to their external appearance [72–74]. Even mild forms of acne, for instance, are associated with a reduced quality of life [75]. The study showed that facial skin impurities were the most common symptom after discontinuation of OCP therapy. Even the mild symptoms reported by women were associated with reduced self-esteem and discomfort. Accordingly, medically harmless facial skin complaints after discontinuation represent a major psychological burden for female YouTubers. The literature also describes that women, on average, experience longer cycles and more variability in cycle length after discontinuation [46,47,76]. This is consistent with the experience of female YouTubers. The OCP has also been questioned and studied for its possible negative association with mental [10,25,61,77] and sexual health [10,62,69]. Research on these topics should still be interpreted with caution, as there is limited consistency in the direction of the evidence [10]. Although it did not seem to be a central theme among the YouTubers, women in our sample reported mood swings and a decrease in libido as side effects of OCPs and their improvement after discontinuation. Interestingly, the prescription of OCPs before young girls have a regular sex life was also criticized. Despite the negative impact of health symptoms after discontinuation, such as facial skin impurities, the women overwhelmingly experienced discontinuation as positive.

When OCPs are discontinued for reasons other than pregnancy, the use of other effective contraceptive methods is very important for reproductive health, otherwise, women are at high risk of becoming pregnant. The data show that most YouTubers switch to condoms or a copper implant (IUD, ball, or chain). However, NFP, a modern fertility awareness-based method, is also used quite frequently. Both copper implants [78] and modern fertility awareness-based methods are effective ways to prevent pregnancy [79,80]. In addition, research shows that women have an increasing desire for more autonomy in their contraceptive choices [81–84]. Informed women's choices are associated with better contraceptive adherence and fewer contraceptive failures [85]. However, there may be differences in the counseling and recommendations of healthcare providers regarding contraceptive methods. Research by Irala

et al. (2011), for example, suggests that OCPs and the IUD were predominantly chosen as contraceptive methods because of the doctor's suggestion/recommendation rather than because of the women's wishes [86]. The opposite was true for the male condom or modern fertility awareness-based methods. A German study (2022) found that participants said they would like to be better informed about the OCP [29], and that women who received information from their gynecologist were more likely to feel well-informed (OR 1.59, CI: 1.10–2.30) than those who received information from the Internet [29]. Healthcare providers should be made more aware of alternative contraceptive methods other than OCPs to increase the autonomy in contraceptive choice and ensure contraceptive effectiveness.

Looking at all the results, the body and bodily perception seem to play a central role in all phases of the OCP autobiography. The embodiment is reflected in the reasons for initiating and weaning off OCPs, the side effects experienced, and the changes that occur after stopping the OCP [79]. This role of bodies and contraception is consistent with other research: the OCP and other contraceptive methods, for example, have led to women having greater control over their reproductive bodies [80,81], and the importance of the (gendered) body in women's contraceptive decision-making [36,80,82].

### Future research directions

The framing of health information can influence health-related beliefs, attitudes, and behaviors [87]. The increase in information and interpretation of scientific evidence in the media can improve informed contraceptive choices. Conversely, they can also negatively affect public health through misinterpretation of scientific evidence [28]. We emphasize the need to further investigate the relationship between SNSs and individual contraceptive method choice discontinuation. There is a need for longitudinal and qualitative research to examine and explore the underlying processes and how these choices change over the life course [42]. In the context of our study design, it would be interesting to investigate if and how the private experiences of influencers on YouTube and other SNSs platforms (e.g. Instagram & TikTok) are used as health information regarding method discontinuation choices. In particular, women who seek contraceptive information online appear to have lower levels of trust in information provided by their gynecologists [29]. It would also be interesting to explore the underlying motivations and phenomena behind the public uploading of such private information, especially within the dynamics of self-presentation and online marketing.

### Strengths and limitations

The methodological strength of this study was its mixed methods design. It allowed for a greater contextual understanding of the descriptive data presented through the qualitative analysis. In addition, the findings represent the personal views and experiences of the women. These are naturally expressed and not biased by the research design itself. However, because these findings are based on YouTube, they may have limited generalizability and transferability to the general population. The sample may be biased because the women who post content on YouTube may be different from the average woman who discontinues hormonal contraception. This recruitment method did not allow us to collect important background information (e.g. socioeconomic status, gender, ethnicity). Furthermore, for a large proportion of our study participants, we had to assume from looks and past upload history that they were adults. The study is also limited in terms of validating the qualitative aspect of this study as the data is retrospective, thus, preventing iteration and triangulation as additional validation methods. In addition, women who experience severe side effects during use and health symptoms after discontinuation may be more likely to post their experiences on YouTube than women with no

side effects or symptoms. Thus, we do not know which of the data is factual and which might be influenced by self-presentation on SNSs. The reach of the videos may also have led women to be less open about this sensitive topic than, for example, in face-to-face interviews. This study is also limited to the content of what women report in their videos. It cannot, for example, consider changes in life circumstances (e.g. change of life/sexual partner, facial skin care routine).

## Conclusions

This content analysis of YouTube videos portrayed the personal implementation and discontinuation experiences of OCPs among female YouTubers. In doing so, the study provided valuable insights into the lived experiences, perceptions, and opinions of women who discontinued OCPs in the context of quality healthcare systems. Future qualitative and quantitative research is needed to provide information on the motivations, subsequent health symptoms, and healthcare needs associated with discontinuing OCPs.

## Supporting information

**S1 Table. German search terms.**
(DOCX)

**S2 Table. Qualitative coding scheme.**
(DOCX)

**S3 Table. Video descriptives, detail (N = 175).**
(DOCX)

**S1 File. Translated codebook for the quantitative content analysis.**
(PDF)

## Acknowledgments

We recognize that people other than cisgender women have menstrual periods and use birth control to prevent pregnancy. In this article, we understand the word "women" to be an inclusive term. However, when referencing research, we use the term used to describe the participants throughout the publication.

We want to thank the thoughtful comments of the reviewers of this study for their encouraging thoughts and comments on this publication.

## Author Contributions

**Conceptualization:** Jana Niemann.

**Data curation:** Jana Niemann.

**Formal analysis:** Jana Niemann, Lea Wicherski, Lisa Glaum.

**Investigation:** Jana Niemann, Lea Wicherski.

**Methodology:** Jana Niemann, Lea Wicherski.

**Project administration:** Jana Niemann.

**Supervision:** Liane Schenk, Getraud Stadler, Matthias Richter.

**Validation:** Jana Niemann.

**Visualization:** Jana Niemann.

**Writing – original draft:** Jana Niemann.

**Writing – review & editing:** Jana Niemann, Lea Wicherski.

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
