## [Decision Letter · Decision Letter 0]

8 Feb 2024

PONE-D-24-01385YouTube and the implementation and discontinuation the oral contraceptive pill: A mixed-method content analysisPLOS ONE

Dear Dr. Niemann

Thank you for submitting your manuscript to PLOS ONE. After careful consideration, we feel that it has merit but does not fully meet PLOS ONE’s publication criteria as it currently stands. Therefore, we invite you to submit a revised version of the manuscript that addresses the points raised during the review process.

**Interesting and innovative research but technically it needs attention. Typographical and syntax errors. Please ask a professional to edit or proofread the document.****Please see the comments below and comments in the attached document.** Please submit your revised manuscript by  Mar 24 2024 11:59PM. If you will need more time than this to complete your revisions, please reply to this message or contact the journal office at plosone@plos.org. Please include the following items when submitting your revised manuscript:A rebuttal letter that responds to each point raised by the academic editor and reviewer(s). You should upload this letter as a separate file labeled 'Response to Reviewers'.A marked-up copy of your manuscript that highlights changes made to the original version. You should upload this as a separate file labeled 'Revised Manuscript with Track Changes'.An unmarked version of your revised paper without tracked changes. You should upload this as a separate file labeled 'Manuscript'.If applicable, we recommend that you deposit your laboratory protocols in protocols.io to enhance the reproducibility of your results. Protocols.io assigns your protocol its own identifier (DOI) so that it can be cited independently in the future. For instructions see: https://journals.plos.org/plosone/s/submission-guidelines#loc-laboratory-protocols. Additionally, PLOS ONE offers an option for publishing peer-reviewed Lab Protocol articles, which describe protocols hosted on protocols.io. Read more information on sharing protocols at https://plos.org/protocols?utm_medium=editorial-email&utm_source=authorletters&utm_campaign=protocols.

We look forward to receiving your revised manuscript.

Kind regards,

Deidre Pretorius, PhD

Academic Editor

PLOS ONE

Journal Requirements:

2. In your Methods section, please include additional information about your dataset and ensure that you have included a statement specifying whether the collection and analysis method complied with the terms and conditions for the source of the data.

Additional Editor Comments (if provided):

Interesting and innovative research however, it needs serious technical attention.

Reviewer 1:

This was a very interesting study exploring OCP discontinuation using social media (Youtube) and I enjoyed reading it. This study demonstrates the utility of using Youtube as an acceptable way to explore the lived experiences of women and their choices on OCP, particularly discontinuation. I have a few comments which I hope will strengthen the paper:

Introduction

p.4 line 37 remove the brackets from mainstream. It is an important concept that described the locality of Youtube in social media and should not be in brackets.

p.4 line 40 Pill-scare - perhaps consider adding the word phenomenon to bring across that it was an observed situation that was in question. Also, add where this took place - was it a global phenomenon? Or mostly in US or Europe = HICs?

p.4 line 43 which countries did those studies on unintended pregnancies and abortion take place?

p.4 line 44 instead of beginning the sentence with 'however', consider using 'while'

p.4 line 45 In Germany, this stance was also taken up... (add the word stance). Also, expand a little more on the pill report for context.

p.4 line 47 ...the media focused on the findings of increased risk of depression...(add the word findings for better coherence of this sentence)

p.4 line 48 consider change the word context to community

p.4 line 49 are the weaknesses that the findings could not be generalised?

p.5 line 55 rephrase this sentence for better clarity

p.5 line 59 remove the brackets from hormonal

p.5 line 69 and 'on these sites' to bring across that the shortcoming were related to these sites

p.5 line 76 Add an in-text citation for the S&R Justice Framework; also expand more on this justice lens and it's relevance to your study

p.6 line 80 comma after reason. In text citation for that sentence as it is a definition (or it is the way the researchers have defined it in the study of drawn from a definition in the literature?). Please indicate what the German study was a bout, In a German study on...

Throughout the document please ensure that your in-text citations are in line with the journal requirements - usually at the end of the sentence, rather than for example next to the authors surname at the beginning of the sentence. This may just be a difference in our styles so ensure it meets the journal requirements.

p.6 line 84 relationships...with? Is it intimate/family/friends?

p.6 line 94 mostly positive = in what kind of ways? Agreement? or shared experiences of discontinuation

p.6 line 103 live video and recordings?

p.7 lines 109 - 116 perhaps consider an overall research question or aim, with these RQs reframed as objectives? It will make it easier to begin that opening paragraph in the discussion a lot more succinct and coherent. See my comment on that later.

METHODS

P.7 line 119 pre-registered - what does pre-reg involve? Was that of the study of the video?

p.7 line 121 were findings interpreted together? Where di the mixing take place? In the abstract the researchers state that this was a concurrent explanatory study - this must come through clearly in the design section of this paper.

p.7 126-127 keep the date format consistent. Also explain why the browser and cookies history were cleared & how this links to algorithms and what shows up in your search results/feed?

p.8 line 135 = what was the relevance filter?

p.8 line 140 If repeated

p.9 the flow chart should have arrows illustrating how the decisions took place

p.10 line 188 registration of?

p.11 line 199 Who did the transcribing?

How could you assess that they are adults? Could this be a limitation?

p.11 line 200 where did you obtain ethical clearance to conduct this study? I understand that informed consent wasn't a requirement (as per lines 202 - 203). Also, please expand on the ethical appropriateness of conducting the study without informed consent from the Youtubers.

p.11 line 208 what do you mean by different media?

RESULTS

p.12 line 226 facial skin = facial skin impurities? please ensure this is consistent throughout to illustrate that it is the impurities that became a problem rather than just facial skin.

p.12 line 229 is it about easing menstrual impurities?

p.12 lines 235 - 236 Is it about irregular periods or a decrease in relation to heavy bleeding?

p.1 line 246 add an example so it is consistent with the previous results

p.13 247 mention % in the table = frequency?

p.14 line 248 double nomination = overlap?p.15 line 268 what is the meaning of extreme in this context of libido?

p.15 line 271 restructure sentence for better coherence = 32 women experienced psychological changes..

p.15 line 272 An increase in facial skin blemishes were..

p.15 line 275 remove exclamation point (!)

p.16 line 293 mention = frequency

p.20 line 315 very interesting please expand a bit more here on "not an issue between a man and woman"

DISCUSSION

p.21 lines 339 - 354 My suggestion is that you start with the overall RQ (I suggested earlier) or aim and then highlight the key findings from the study. Be more coherent in this opening paragraph of the discussion. At the moment, it is written so that it overlaps with the results section.

p.22 lines 372 The main reported reasons...

p.23 line 391 caution in interpretation = why?

p.23 line 408 in-text citation is missing here and you're citing a German study.

p.24 line 424 ...method choice, particularly discontinuation. Remove brackets.

p.24 line 425 add how these choices change over the life course.

p.24 line 431 ...information publicly, especially in the context (merge the sentences)

p.24 line 434 please emphasise where the mixing of methods took place?

Reviewer 2

This study was an innovative method for data collection as it avoided ethical considerations as well as labour-intensive participant recruitment.

It does, however, have some limitations in terms of the validation methods for the qualitative aspect of this study. Because the data is retrospective, iteration was not possible. It was also not possible for triangulation as another method to validate your results. Perhaps self-reflection (reflexivity) should also be considered. These are just aspects to acknowledge. I do not think it disqualifies this study from being published.

There are some typing errors and many grammatical problems. Many sentences are difficult to understand. I would suggest proofreading and language editing to be done by a certified professional before resubmission. I have highlighted a few typing errors and confusion re meaning of terms or sentences, but my role was not to proofread, or language edit this manuscript as other professionals should assist you with it.

Reviewers' comments:

Reviewer's Responses to Questions

**Comments to the Author**

1. Is the manuscript technically sound, and do the data support the conclusions?

Reviewer #1: Yes

Reviewer #2: Yes

2. Has the statistical analysis been performed appropriately and rigorously? 

Reviewer #1: Yes

Reviewer #2: Yes

3. Have the authors made all data underlying the findings in their manuscript fully available?

Reviewer #1: Yes

Reviewer #2: Yes

4. Is the manuscript presented in an intelligible fashion and written in standard English?

Reviewer #1: Yes

Reviewer #2: No

5. Review Comments to the Author

Reviewer #1: This was a very interesting study exploring OCP discontinuation using social media (Youtube) and I enjoyed reading it. This study demonstrates the utility of using Youtube as an acceptable way to explore the lived experiences of women and their choices on OCP, particularly discontinuation. I have a few comments which I hope will strengthen the paper:

Introduction

p.4 line 37 remove the brackets from mainstream. It is an important concept that described the locality of Youtube in social media and should not be in brackets.

p.4 line 40 Pill-scare - perhaps consider adding the word phenomenon to bring across that it was an observed situation that was in question. Also, add where this took place - was it a global phenomenon? Or mostly in US or Europe = HICs?

p.4 line 43 which countries did those studies on unintended pregnancies and abortion take place?

p.4 line 44 instead of beginning the sentence with 'however', consider using 'while'

p.4 line 45 In Germany, this stance was also taken up... (add the word stance). Also, expand a little more on the pill report for context.

p.4 line 47 ...the media focused on the findings of increased risk of depression...(add the word findings for better coherence of this sentence)

p.4 line 48 consider change the word context to community

p.4 line 49 are the weaknesses that the findings could not be generalised?

p.5 line 55 rephrase this sentence for better clarity

p.5 line 59 remove the brackets from hormonal

p.5 line 69 and 'on these sites' to bring across that the shortcoming were related to these sites

p.5 line 76 Add an in-text citation for the S&R Justice Framework; also expand more on this justice lens and it's relevance to your study

p.6 line 80 comma after reason. In text citation for that sentence as it is a definition (or it is the way the researchers have defined it in the study of drawn from a definition in the literature?). Please indicate what the German study was a bout, In a German study on...

Throughout the document please ensure that your in-text citations are in line with the journal requirements - usually at the end of the sentence, rather than for example next to the authors surname at the beginning of the sentence. This may just be a difference in our styles so ensure it meets the journal requirements.

p.6 line 84 relationships...with? Is it intimate/family/friends?

p.6 line 94 mostly positive = in what kind of ways? Agreement? or shared experiences of discontinuation

p.6 line 103 live video and recordings?

p.7 lines 109 - 116 perhaps consider an overall research question or aim, with these RQs reframed as objectives? It will make it easier to begin that opening paragraph in the discussion a lot more succinct and coherent. See my comment on that later.

METHODS

P.7 line 119 pre-registered - what does pre-reg involve? Was that of the study of the video?

p.7 line 121 were findings interpreted together? Where di the mixing take place? In the abstract the researchers state that this was a concurrent explanatory study - this must come through clearly in the design section of this paper.

p.7 126-127 keep the date format consistent. Also explain why the browser and cookies history were cleared & how this links to algorithms and what shows up in your search results/feed?

p.8 line 135 = what was the relevance filter?

p.8 line 140 If repeated

p.9 the flow chart should have arrows illustrating how the decisions took place

p.10 line 188 registration of?

p.11 line 199 Who did the transcribing?

How could you assess that they are adults? Could this be a limitation?

p.11 line 200 where did you obtain ethical clearance to conduct this study? I understand that informed consent wasn't a requirement (as per lines 202 - 203). Also, please expand on the ethical appropriateness of conducting the study without informed consent from the Youtubers.

p.11 line 208 what do you mean by different media?

RESULTS

p.12 line 226 facial skin = facial skin impurities? please ensure this is consistent throughout to illustrate that it is the impurities that became a problem rather than just facial skin.

p.12 line 229 is it about easing menstrual impurities?

p.12 lines 235 - 236 Is it about irregular periods or a decrease in relation to heavy bleeding?

p.1 line 246 add an example so it is consistent with the previous results

p.13 247 mention % in the table = frequency?

p.14 lin 248 double nomination = overlap?p.15 line 268 what is th emeaning of extreme in this context of libido?

p.15 line 271 restructure sentence for better coherence = 32 women experienced psychological changes..

p.15 line 272 An increase in facial skin blemishes were..

p.15 line 275 remove exclamation point (!)

p.16 line 293 mention = frequency

p.20 line 315 very interesting please expand a bit more here on "not an issue between a man and woman"

DISCUSSION

p.21 lines 339 - 354 My suggestion is that you start with the overall RQ (I suggested earlier) or aim and then highlight the key findings from the study. Be more coherent in this opening paragraph of the discussion. At the moment, it is written so that it overlaps with the results section.

p.22 lines 372 The main reported reasons...

p.23 line 391 caution in interpretation = why?

p.23 line 408 in-text citation is missing here and you're citing a German study.

p.24 line 424 ...method choice, particularly discontinuation. Remove brackets.

p.24 line 425 add how these choices change over the life course.

p.24 line 431 ...information publicly, especially in the context (merge the sentences)

p.24 line 434 please emphasise where the mixing of methods took place?

Reviewer #2: This study was an innovative method for data collection as it avoided ethical considerations as well as labour-intensive participant recruitment.

It does, however, have some limitations in terms of the validation methods for the qualitative aspect of this study. Because the data is retrospective, iteration was not possible. It was also not possible for triangulation as another method to validate your results. Perhaps self reflection (reflexivity) should also be considered. These are just aspects to acknowledge. I do not think it disqualifies this study from being published.

There are some typing errors and many grammatical problems. Many sentences are difficult to understand. I would suggest proofreading and language editing to be done by a certified professional before resubmission. I have highlighted a few typing errors and confusion re meaning of terms or sentences, but my role was not to proofread or language edit this manuscript as other professionals should assist you with it.

6. PLOS authors have the option to publish the peer review history of their article (what does this mean?). If published, this will include your full peer review and any attached files.

Reviewer #1: **Yes: **Janine White

Reviewer #2: No

---

## [Author Response · Author response to Decision Letter 0]

8 Mar 2024

A point-by-point response to reviewers

Dear Dr Deidre Pretorius, dear reviewers,

Thank you very much for your thoughtful comments and for the opportunity to resubmit our article. We are grateful that you are considering publishing it in PLosOne.

Please find below our responses to your comments and the reviewers’ comments. Our re-sponses are highlighted in italics. We have marked changes to the previous version in red font in the marked-up copy of the manuscript.

We are looking forward to your feedback.

With kind regards,

Jana Niemann, Lea Wicherski, Lisa Glaum, Liane Schenk, Getraud Stadler, Matthias Richter

Journal Requirements

Authors: We have adjusted the manuscript format to PLOS ONE’s style requirements. 

2. In your Methods section, please include additional information about your dataset and en-sure that you have included a statement specifying whether the collection and analysis meth-od complied with the terms and conditions for the source of the data.

Authors: We have added the following statement to the end of the ethics section: “The collec-tion and analysis method complied with the terms and conditions for the source of the data.”

Please confirm at this time whether or not your submission contains all raw data required to replicate the results of your study. Authors must share the “minimal data set” for their sub-mission. PLOS defines the minimal data set to consist of the data required to replicate all study findings reported in the article, as well as related metadata and methods (https://journals.plos.org/plosone/s/data-availability#loc-minimal-data-set-definition).

If your submission does not contain these data, please either upload them as Supporting In-formation files or deposit them to a stable, public repository and provide us with the relevant URLs, DOIs, or accession numbers. For a list of recommended repositories, please see https://journals.plos.org/plosone/s/recommended-repositories.

If there are ethical or legal restrictions on sharing a de-identified data set, please explain them in detail (e.g., data contain potentially sensitive information, data are owned by a third-party organization, etc.) and who has imposed them (e.g., an ethics committee). Please also pro-vide contact information for a data access committee, ethics committee, or other institutional body to which data requests may be sent. If data are owned by a third party, please indicate how others may request data access.

Authors: With this revision, we have added the raw data of the quantitative data to our OSF project and added the following sentence: 

“The study follows the Open Science movement, i.e. the pre-registration, and all quantitative data are stored on the Open Science Foundation server (https://osf.io/fekdh/).”

For the qualitative research, we would not feel comfortable uploading the raw transcripts with the manuscript. We want to provide some privacy to the participants in the study.

Authors: We have gone through all references of the revised manuscript. We have found no retracted articles. We have added the following references to accompany the revision: 

• 40. Ross LJ. Reproductive Justice as Intersectional Feminist Activism. Souls. 2017;19: 286–314. doi:10.1080/10999949.2017.1389634

• 41. Castle S, Askew I, Population Council. CONTRACEPTIVE DISCONTINUATION: REASONS, CHALLENGES, AND SOLUTIONS. 2015. Available: https://popdesenvolvimento.org/images/imprensa/FP2020_ContraceptiveDiscontinuation_SinglePageRevise_12.16.15.pdf

• 60.Patterson AN. YouTube Generated Video Clips as Qualitative Research Data: One Researcher’s Reflections on the Process. Qual Inq. 2018;24: 759–767. doi:10.1177/1077800418788107

We have deleted one citation from the text, as we cited it incorrectly: 

• Ekrami O, Claes P, White JD, Zaidi AA, Shriver MD, Van Dongen S. Measuring asymmetry from high-density 3D surface scans: An application to human faces. PLoS One. 2018;13. doi:10.1371/journal.pone.0207895

We have added the DOI to the following citation: 

• 43. Gnoth C, Frank-Herrmann P, Schmoll A, Godehardt E, Freundl G. Cycle characteristics after discontinuation of oral contraceptives. Gynecol Endocrinol. 2002;16: 307–317. doi:10.1080/gye.16.4.307.317

We have added the URL to the following citation: 

• 64. Hardon A, Harries J. Towards an Anthropology of contraception: on the pill, control and embodiment. AM Riv Della Soc Ital Di Antropol Medica. 2001;6: 211–226. Availa-ble: https://www.amantropologiamedica.unipg.it/index.php/am/article/view/119/112

We have adapted the title in the following citation:

• 67. Tracy EE. Contraception: Menarche to Menopause. Obstet Gynecol Clin North Am. 2017;44: 143–158. doi:10.1016/j.ogc.2017.02.001

Finally, we have adapted the following citation: 

• 87. Gallagher KM, Updegraff JA. Health Message Framing Effects on Attitudes, Inten-tions, and Behavior: A Meta-analytic Review. Ann Behav Med. 2012;43: 101–116. doi:10.1007/s12160-011-9308-7

Editor

Interesting and innovative research however, it needs serious technical attention.

Authors: Thank you for your feedback. We have gone through the entire manuscript with an eye for technical errors. We also hired a professional to proofread the manuscript.

Reviewer 1 - Janine White

This was a very interesting study exploring OCP discontinuation using social media (Youtube) and I enjoyed reading it. This study demonstrates the utility of using Youtube as an acceptable way to explore the lived experiences of women and their choices on OCP, particularly discon-tinuation. I have a few comments which I hope will strengthen the paper:

Introduction

Authors: Dear Janine White, thank you for your encouraging remarks and for challenging us to improve the paper. We detail below whether and how we addressed your comments.

p.4 line 37 remove the brackets from mainstream. It is an important concept that de-scribed the locality of Youtube in social media and should not be in brackets.

Authors: We have adjusted the text passage accordingly.

p.4 line 40 Pill-scare - perhaps consider adding the word phenomenon to bring across that it was an observed situation that was in question. Also, add where this took place - was it a global phenomenon? Or mostly in US or Europe = HICs?

Authors: Thank you very much. We changed the wording of the sentences to “An im-portant example is the “pill scare” phenomenon of 1995 [16–18], focusing on high-income countries.” (see lines 41-42).

p.4 line 43 which countries did those studies on unintended pregnancies and abortion take place?

Authors: We have added the countries to the references. 

p.4 line 44 instead of beginning the sentence with 'however', consider using 'while'

Authors: We have adjusted the text passage accordingly.

p.4 line 45 In Germany, this stance was also taken up... (add the word stance). Also, expand a little more on the pill report for context.

Authors: We have added “stance” and more information on the pill report: “This stance was also taken up in the OCP report [11] in Germany in 2015, which lacked scientific rigor [19]. The authors concluded there was a higher risk of thrombosis with the more mod-ern 3rd and 4th generation OCPs. This was further disseminated by the media [20–22].” (lines 47-49)

p.4 line 47 ...the media focused on the findings of increased risk of depression...(add the word findings for better coherence of this sentence)

Authors: We have added the word “findings” accordingly.

p.4 line 48 consider change the word context to community

Authors: We have exchanged the word accordingly.

p.4 line 49 are the weaknesses that the findings could not be generalised?

Authors: We have added the following sentence to give examples of the criticism: “Bitzer (2017), for example, criticized the lack of sensitivity analysis and questioned the biological plausibility [24].” (lines 52-53)

p.5 line 55 rephrase this sentence for better clarity

Authors: We have reworded the sentence and hope that it is now clearer: “In sum-mary, the media discussion of OCPs has underrepresented studies that do not show cor-relations with adverse side effects and have often failed to distinguish between absolute and rela-tive risks [15,23].” (lines 59-60)

p.5 line 59 remove the brackets from hormonal

Authors: We have removed the brackets. 

p.5 line 69 and 'on these sites' to bring across that the shortcoming were related to these sites

Authors: We have added the word “on these sites” accordingly.

p.5 line 76 Add an in-text citation for the S&R Justice Framework; also expand more on this justice lens and it's relevance to your study

Authors: We added an in-text citation and expanded on the justice lens regarding con-traceptive agency and contraceptive choice: “Informed choice regarding contraceptive methods is an important component of the Sexual and Reproductive Justice Frame-work [26,40]. Knowledge and information can contribute to achieving justice through increased reproductive agency and contraceptive choice [26].” (lines 79-81)

p.6 line 80 comma after reason. In text citation for that sentence as it is a definition (or it is the way the researchers have defined it in the study of drawn from a definition in the literature?). Please indicate what the German study was a bout, In a German study on...

Authors: We added the comma to the sentence and indicated what the German study was about. We also added a direct quote from a similar definition of discontinuation with a reference: “The OCP can be defined as “starting contraceptive use and then stopping for any reason while still at risk of an unintended pregnancy” [41]. A German study on the knowledge and perceptions about OCPs among young women showed that 64.5 % (n = 1480) of those who used OCPs at some point discontinued the meth-od [29].” (lines 84-87)

Throughout the document please ensure that your in-text citations are in line with the journal requirements - usually at the end of the sentence, rather than for example next to the authors surname at the beginning of the sentence. This may just be a difference in our styles so ensure it meets the journal requirements.

Authors: We have adjusted the citations accordingly.

p.6 line 84 relationships...with? Is it intimate/family/friends?

Authors: We have added examples for the possible relationships. 

p.6 line 94 mostly positive = in what kind of ways? Agreement? or shared experiences of discontinuation

Authors: We have added that the reactions were “very positive with many views and likes, very few dislikes and lots of friendly comments.” (lines 100-101)

p.6 line 103 live video and recordings?

Authors: We have deleted the word “live” from the sentence”

p.7 lines 109 - 116 perhaps consider an overall research question or aim, with these RQs reframed as objectives? It will make it easier to begin that opening paragraph in the discussion a lot more succinct and coherent. See my comment on that later.

Authors: We have rewritten the objectives and research questions as you suggested. We hope that the paragraph and the beginning of the discussion are clearer now: “The overall objective of this content analysis was to investigate the personal physiological and psychological changes and lived experiences of German-speaking YouTubers af-ter initiation and discontinuation of OCP treatment. We, therefore, aim to examine

• the reasons for starting and discontinuing the use of the OCP, 

• to document any side effects experienced during and after use, and

• how the women describe their history with the OCP.” (lines 114-119)

METHODS

P.7 line 119 pre-registered - what does pre-reg involve? Was that of the study of the video?

Authors: We have clarified that we registered a protocol for this study and restructured the beginning of the paragraph as follows: “The study follows the Open Science movement, i.e. the pre-registration, and all data are stored on the Open Science Foundation server (https://osf.io/fekdh/). It was designed as an (almost) simultaneous exploratory mixed-methods content analysis [52]” (lines 122-124).

p.7 line 121 were findings interpreted together? Where di the mixing take place? In the abstract the researchers state that this was a concurrent explanatory study - this must come through clearly in the design section of this paper.

Authors: We have rearranged the paragraph of the study design and added that: “The qualitative and quantitative research strands were initially analyzed separately and then brought together for the interpretation and presentation of the results.” (lines 127-128).

p.7 126-127 keep the date format consistent. Also explain why the browser and cook-ies history were cleared & how this links to algorithms and what shows up in your search results/feed?

Authors: We’ve changed the date format and explained why we cleared the browser and cookies history: “YouTube was initially searched from July 20 to 23, 2021. An up-dated search was performed on May 5, 2023. The internet browser and cookies histo-ry were cleared before the search to avoid biases from our laptop.” (lines 130-132)

p.8 line 135 = what was the relevance filter?

Authors: We have added the relevance filter (=year) to the sentence

p.8 line 140 If repeated

Authors: We considered your suggestion and adapted the sentence.

p.9 the flow chart should have arrows illustrating how the decisions took place

Authors: We are not sure what you are referring to. We have based this flowchart on the PRISMA flowchart to illustrate the decision process. The flowchart includes arrows to indicate direction and how the boxes are linked.

p.10 line 188 registration of?

Authors: We have deleted the “registration of” from the sentence. 

p.11 line 199 Who did the transcribing?

Authors: We have added the authors to the sentence. 

How could you assess that they are adults? Could this be a limitation?

Authors: Yes, this is indeed a limitation. For more than half of the participants, we know the age at which they stopped. For the rest of the enrolled participants, we only as-sumed they were adults based on appearance and previous upload history.

We have added this limitation to the strengths and limitations section.: “Furthermore, for a large proportion of our study participants, we had to assume from looks and past upload history that they were adults.” (lines 464-465)

p.11 line 200 where did you obtain ethical clearance to conduct this study? I under-stand that informed consent wasn't a requirement (as per lines 202 - 203). Also, please expand on the ethical appropriateness of conducting the study without informed consent from the Youtubers.

Authors: We have added our ethical approval for the study. In addition, we have further considered the ethical appropriateness of our data: “The ethical approval for this study has been granted by the Ethical Review Committee of the Medical Faculty at Martin Luther University Halle-Wittenberg.” (lines 211- 212).

And have extended a little bit on the ethical appropriateness of conducting the study: “Before a person uploads a video on YouTube, they have to c

---

## [Editor Report · Decision Letter 1]

15 Mar 2024

PONE-D-24-01385R1YouTube and the implementation and discontinuation of the oral contraceptive pill: A mixed-method content analysisPLOS ONE

Dear Dr. Niemann,

Thank you for submitting your manuscript to PLOS ONE. After careful consideration, we feel that it has merit but does not fully meet PLOS ONE’s publication criteria as it currently stands. Therefore, we invite you to submit a revised version of the manuscript that addresses the points raised during the review process.

**Please see the feedback from the reviewers. Please do all the corrections and submit a document indicating the corrections in track changes or motivate why it could not be done.**

We look forward to receiving your revised manuscript.

Kind regards,

Deidre Pretorius, PhD

Academic Editor

PLOS ONE

Journal Requirements:

Additional Editor Comments:

Reviewer 1

This was a very interesting study exploring OCP discontinuation using social media (Youtube) and I enjoyed reading it. This study demonstrates the utility of using Youtube as an acceptable way to explore the lived experiences of women and their choices on OCP, particularly discontinuation. I have a few comments which I hope will strengthen the paper:

Introduction

p.4 line 37 remove the brackets from mainstream. It is an important concept that described the locality of Youtube in social media and should not be in brackets.

p.4 line 40 Pill-scare - perhaps consider adding the word phenomenon to bring across that it was an observed situation that was in question. Also, add where this took place - was it a global phenomenon? Or mostly in US or Europe = HICs?

p.4 line 43 which countries did those studies on unintended pregnancies and abortion take place?

p.4 line 44 instead of beginning the sentence with 'however', consider using 'while'

p.4 line 45 In Germany, this stance was also taken up... (add the word stance). Also, expand a little more on the pill report for context.

p.4 line 47 ...the media focused on the findings of increased risk of depression...(add the word findings for better coherence of this sentence)

p.4 line 48 consider change the word context to community

p.4 line 49 are the weaknesses that the findings could not be generalised?

p.5 line 55 rephrase this sentence for better clarity

p.5 line 59 remove the brackets from hormonal

p.5 line 69 and 'on these sites' to bring across that the shortcoming were related to these sites

p.5 line 76 Add an in-text citation for the S&R Justice Framework; also expand more on this justice lens and it's relevance to your study

p.6 line 80 comma after reason. In text citation for that sentence as it is a definition (or it is the way the researchers have defined it in the study of drawn from a definition in the literature?). Please indicate what the German study was a bout, In a German study on...

Throughout the document please ensure that your in-text citations are in line with the journal requirements - usually at the end of the sentence, rather than for example next to the authors surname at the beginning of the sentence. This may just be a difference in our styles so ensure it meets the journal requirements.

p.6 line 84 relationships...with? Is it intimate/family/friends?

p.6 line 94 mostly positive = in what kind of ways? Agreement? or shared experiences of discontinuation

p.6 line 103 live video and recordings?

p.7 lines 109 - 116 perhaps consider an overall research question or aim, with these RQs reframed as objectives? It will make it easier to begin that opening paragraph in the discussion a lot more succinct and coherent. See my comment on that later.

METHODS

P.7 line 119 pre-registered - what does pre-reg involve? Was that of the study of the video?

p.7 line 121 were findings interpreted together? Where di the mixing take place? In the abstract the researchers state that this was a concurrent explanatory study - this must come through clearly in the design section of this paper.

p.7 126-127 keep the date format consistent. Also explain why the browser and cookies history were cleared & how this links to algorithms and what shows up in your search results/feed?

p.8 line 135 = what was the relevance filter?

p.8 line 140 If repeated

p.9 the flow chart should have arrows illustrating how the decisions took place

p.10 line 188 registration of?

p.11 line 199 Who did the transcribing?

How could you assess that they are adults? Could this be a limitation?

p.11 line 200 where did you obtain ethical clearance to conduct this study? I understand that informed consent wasn't a requirement (as per lines 202 - 203). Also, please expand on the ethical appropriateness of conducting the study without informed consent from the Youtubers.

p.11 line 208 what do you mean by different media?

RESULTS

p.12 line 226 facial skin = facial skin impurities? please ensure this is consistent throughout to illustrate that it is the impurities that became a problem rather than just facial skin.

p.12 line 229 is it about easing menstrual impurities?

p.12 lines 235 - 236 Is it about irregular periods or a decrease in relation to heavy bleeding?

p.1 line 246 add an example so it is consistent with the previous results

p.13 247 mention % in the table = frequency?

p.14 lin 248 double nomination = overlap?p.15 line 268 what is th emeaning of extreme in this context of libido?

p.15 line 271 restructure sentence for better coherence = 32 women experienced psychological changes..

p.15 line 272 An increase in facial skin blemishes were..

p.15 line 275 remove exclamation point (!)

p.16 line 293 mention = frequency

p.20 line 315 very interesting please expand a bit more here on "not an issue between a man and woman"

DISCUSSION

p.21 lines 339 - 354 My suggestion is that you start with the overall RQ (I suggested earlier) or aim and then highlight the key findings from the study. Be more coherent in this opening paragraph of the discussion. At the moment, it is written so that it overlaps with the results section.

p.22 lines 372 The main reported reasons...

p.23 line 391 caution in interpretation = why?

p.23 line 408 in-text citation is missing here and you're citing a German study.

p.24 line 424 ...method choice, particularly discontinuation. Remove brackets.

p.24 line 425 add how these choices change over the life course.

p.24 line 431 ...information publicly, especially in the context (merge the sentences)

p.24 line 434 please emphasise where the mixing of methods took place?

REVIEWER 2

This study was an innovative method for data collection as it avoided ethical considerations as well as labour-intensive participant recruitment.

It does, however, have some limitations in terms of the validation methods for the qualitative aspect of this study. Because the data is retrospective, iteration was not possible. It was also not possible for triangulation as another method to validate your results. Perhaps self reflection (reflexivity) should also be considered. These are just aspects to acknowledge. I do not think it disqualifies this study from being published.

There are some typing errors and many grammatical problems. Many sentences are difficult to understand. I would suggest proofreading and language editing to be done by a certified professional before resubmission. I have highlighted a few typing errors and confusion re meaning of terms or sentences, but my role was not to proofread or language edit this manuscript as other professionals should assist you with it.

---

## [Author Response · Author response to Decision Letter 1]

20 Mar 2024

A point-by-point response to reviewers

Dear Dr Deidre Pretorius, dear reviewers,

Thank you very much for your thoughtful comments and for the opportunity to resubmit our article. We are grateful that you are considering publishing it in PLosOne.

Please find below our responses to your comments and the reviewers’ comments. Our re-sponses are highlighted in italics. We have marked changes to the previous version in red font in the marked-up copy of the manuscript.

We are looking forward to your feedback.

With kind regards,

Jana Niemann, Lea Wicherski, Lisa Glaum, Liane Schenk, Getraud Stadler, Matthias Richter

Journal Requirements

Authors: We have adjusted the manuscript format to PLOS ONE’s style requirements. 

2. In your Methods section, please include additional information about your dataset and en-sure that you have included a statement specifying whether the collection and analysis meth-od complied with the terms and conditions for the source of the data.

Authors: We have added the following statement to the end of the ethics section: “The collec-tion and analysis method complied with the terms and conditions for the source of the data.”

Please confirm at this time whether or not your submission contains all raw data required to replicate the results of your study. Authors must share the “minimal data set” for their sub-mission. PLOS defines the minimal data set to consist of the data required to replicate all study findings reported in the article, as well as related metadata and methods (https://journals.plos.org/plosone/s/data-availability#loc-minimal-data-set-definition).

If your submission does not contain these data, please either upload them as Supporting In-formation files or deposit them to a stable, public repository and provide us with the relevant URLs, DOIs, or accession numbers. For a list of recommended repositories, please see https://journals.plos.org/plosone/s/recommended-repositories.

If there are ethical or legal restrictions on sharing a de-identified data set, please explain them in detail (e.g., data contain potentially sensitive information, data are owned by a third-party organization, etc.) and who has imposed them (e.g., an ethics committee). Please also pro-vide contact information for a data access committee, ethics committee, or other institutional body to which data requests may be sent. If data are owned by a third party, please indicate how others may request data access.

Authors: With this revision, we have added the raw data of the quantitative data to our OSF project and added the following sentence: 

“The study follows the Open Science movement, i.e. the pre-registration, and all quantitative data are stored on the Open Science Foundation server (https://osf.io/fekdh/).”

For the qualitative research, we would not feel comfortable uploading the raw transcripts with the manuscript. We want to provide some privacy to the participants in the study.

Authors: We have gone through all references of the revised manuscript. We have found no retracted articles. We have added the following references to accompany the revision: 

• 40. Ross LJ. Reproductive Justice as Intersectional Feminist Activism. Souls. 2017;19: 286–314. doi:10.1080/10999949.2017.1389634

• 41. Castle S, Askew I, Population Council. CONTRACEPTIVE DISCONTINUATION: REASONS, CHALLENGES, AND SOLUTIONS. 2015. Available: https://popdesenvolvimento.org/images/imprensa/FP2020_ContraceptiveDiscontinuation_SinglePageRevise_12.16.15.pdf

• 60.Patterson AN. YouTube Generated Video Clips as Qualitative Research Data: One Researcher’s Reflections on the Process. Qual Inq. 2018;24: 759–767. doi:10.1177/1077800418788107

We have deleted one citation from the text, as we cited it incorrectly: 

• Ekrami O, Claes P, White JD, Zaidi AA, Shriver MD, Van Dongen S. Measuring asymmetry from high-density 3D surface scans: An application to human faces. PLoS One. 2018;13. doi:10.1371/journal.pone.0207895

We have added the DOI to the following citation: 

• 43. Gnoth C, Frank-Herrmann P, Schmoll A, Godehardt E, Freundl G. Cycle characteristics after discontinuation of oral contraceptives. Gynecol Endocrinol. 2002;16: 307–317. doi:10.1080/gye.16.4.307.317

We have added the URL to the following citation: 

• 64. Hardon A, Harries J. Towards an Anthropology of contraception: on the pill, control and embodiment. AM Riv Della Soc Ital Di Antropol Medica. 2001;6: 211–226. Availa-ble: https://www.amantropologiamedica.unipg.it/index.php/am/article/view/119/112

We have adapted the title in the following citation:

• 67. Tracy EE. Contraception: Menarche to Menopause. Obstet Gynecol Clin North Am. 2017;44: 143–158. doi:10.1016/j.ogc.2017.02.001

Finally, we have adapted the following citation: 

• 87. Gallagher KM, Updegraff JA. Health Message Framing Effects on Attitudes, Inten-tions, and Behavior: A Meta-analytic Review. Ann Behav Med. 2012;43: 101–116. doi:10.1007/s12160-011-9308-7

Editor

Interesting and innovative research however, it needs serious technical attention.

Authors: Thank you for your feedback. We have gone through the entire manuscript with an eye for technical errors. We also hired a professional to proofread the manuscript.

Reviewer 1 - Janine White

This was a very interesting study exploring OCP discontinuation using social media (Youtube) and I enjoyed reading it. This study demonstrates the utility of using Youtube as an acceptable way to explore the lived experiences of women and their choices on OCP, particularly discon-tinuation. I have a few comments which I hope will strengthen the paper:

Introduction

Authors: Dear Janine White, thank you for your encouraging remarks and for challenging us to improve the paper. We detail below whether and how we addressed your comments.

p.4 line 37 remove the brackets from mainstream. It is an important concept that de-scribed the locality of Youtube in social media and should not be in brackets.

Authors: We have adjusted the text passage accordingly.

p.4 line 40 Pill-scare - perhaps consider adding the word phenomenon to bring across that it was an observed situation that was in question. Also, add where this took place - was it a global phenomenon? Or mostly in US or Europe = HICs?

Authors: Thank you very much. We changed the wording of the sentences to “An im-portant example is the “pill scare” phenomenon of 1995 [16–18], focusing on high-income countries.” (see lines 41-42).

p.4 line 43 which countries did those studies on unintended pregnancies and abortion take place?

Authors: We have added the countries to the references. 

p.4 line 44 instead of beginning the sentence with 'however', consider using 'while'

Authors: We have adjusted the text passage accordingly.

p.4 line 45 In Germany, this stance was also taken up... (add the word stance). Also, expand a little more on the pill report for context.

Authors: We have added “stance” and more information on the pill report: “This stance was also taken up in the OCP report [11] in Germany in 2015, which lacked scientific rigor [19]. The authors concluded there was a higher risk of thrombosis with the more mod-ern 3rd and 4th generation OCPs. This was further disseminated by the media [20–22].” (lines 47-49)

p.4 line 47 ...the media focused on the findings of increased risk of depression...(add the word findings for better coherence of this sentence)

Authors: We have added the word “findings” accordingly.

p.4 line 48 consider change the word context to community

Authors: We have exchanged the word accordingly.

p.4 line 49 are the weaknesses that the findings could not be generalised?

Authors: We have added the following sentence to give examples of the criticism: “Bitzer (2017), for example, criticized the lack of sensitivity analysis and questioned the biological plausibility [24].” (lines 52-53)

p.5 line 55 rephrase this sentence for better clarity

Authors: We have reworded the sentence and hope that it is now clearer: “In sum-mary, the media discussion of OCPs has underrepresented studies that do not show cor-relations with adverse side effects and have often failed to distinguish between absolute and rela-tive risks [15,23].” (lines 59-60)

p.5 line 59 remove the brackets from hormonal

Authors: We have removed the brackets. 

p.5 line 69 and 'on these sites' to bring across that the shortcoming were related to these sites

Authors: We have added the word “on these sites” accordingly.

p.5 line 76 Add an in-text citation for the S&R Justice Framework; also expand more on this justice lens and it's relevance to your study

Authors: We added an in-text citation and expanded on the justice lens regarding con-traceptive agency and contraceptive choice: “Informed choice regarding contraceptive methods is an important component of the Sexual and Reproductive Justice Frame-work [26,40]. Knowledge and information can contribute to achieving justice through increased reproductive agency and contraceptive choice [26].” (lines 79-81)

p.6 line 80 comma after reason. In text citation for that sentence as it is a definition (or it is the way the researchers have defined it in the study of drawn from a definition in the literature?). Please indicate what the German study was a bout, In a German study on...

Authors: We added the comma to the sentence and indicated what the German study was about. We also added a direct quote from a similar definition of discontinuation with a reference: “The OCP can be defined as “starting contraceptive use and then stopping for any reason while still at risk of an unintended pregnancy” [41]. A German study on the knowledge and perceptions about OCPs among young women showed that 64.5 % (n = 1480) of those who used OCPs at some point discontinued the meth-od [29].” (lines 84-87)

Throughout the document please ensure that your in-text citations are in line with the journal requirements - usually at the end of the sentence, rather than for example next to the authors surname at the beginning of the sentence. This may just be a difference in our styles so ensure it meets the journal requirements.

Authors: We have adjusted the citations accordingly.

p.6 line 84 relationships...with? Is it intimate/family/friends?

Authors: We have added examples for the possible relationships. 

p.6 line 94 mostly positive = in what kind of ways? Agreement? or shared experiences of discontinuation

Authors: We have added that the reactions were “very positive with many views and likes, very few dislikes and lots of friendly comments.” (lines 100-101)

p.6 line 103 live video and recordings?

Authors: We have deleted the word “live” from the sentence”

p.7 lines 109 - 116 perhaps consider an overall research question or aim, with these RQs reframed as objectives? It will make it easier to begin that opening paragraph in the discussion a lot more succinct and coherent. See my comment on that later.

Authors: We have rewritten the objectives and research questions as you suggested. We hope that the paragraph and the beginning of the discussion are clearer now: “The overall objective of this content analysis was to investigate the personal physiological and psychological changes and lived experiences of German-speaking YouTubers af-ter initiation and discontinuation of OCP treatment. We, therefore, aim to examine

• the reasons for starting and discontinuing the use of the OCP, 

• to document any side effects experienced during and after use, and

• how the women describe their history with the OCP.” (lines 114-119)

METHODS

P.7 line 119 pre-registered - what does pre-reg involve? Was that of the study of the video?

Authors: We have clarified that we registered a protocol for this study and restructured the beginning of the paragraph as follows: “The study follows the Open Science movement, i.e. the pre-registration, and all data are stored on the Open Science Foundation server (https://osf.io/fekdh/). It was designed as an (almost) simultaneous exploratory mixed-methods content analysis [52]” (lines 122-124).

p.7 line 121 were findings interpreted together? Where di the mixing take place? In the abstract the researchers state that this was a concurrent explanatory study - this must come through clearly in the design section of this paper.

Authors: We have rearranged the paragraph of the study design and added that: “The qualitative and quantitative research strands were initially analyzed separately and then brought together for the interpretation and presentation of the results.” (lines 127-128).

p.7 126-127 keep the date format consistent. Also explain why the browser and cook-ies history were cleared & how this links to algorithms and what shows up in your search results/feed?

Authors: We’ve changed the date format and explained why we cleared the browser and cookies history: “YouTube was initially searched from July 20 to 23, 2021. An up-dated search was performed on May 5, 2023. The internet browser and cookies histo-ry were cleared before the search to avoid biases from our laptop.” (lines 130-132)

p.8 line 135 = what was the relevance filter?

Authors: We have added the relevance filter (=year) to the sentence

p.8 line 140 If repeated

Authors: We considered your suggestion and adapted the sentence.

p.9 the flow chart should have arrows illustrating how the decisions took place

Authors: We are not sure what you are referring to. We have based this flowchart on the PRISMA flowchart to illustrate the decision process. The flowchart includes arrows to indicate direction and how the boxes are linked.

p.10 line 188 registration of?

Authors: We have deleted the “registration of” from the sentence. 

p.11 line 199 Who did the transcribing?

Authors: We have added the authors to the sentence. 

How could you assess that they are adults? Could this be a limitation?

Authors: Yes, this is indeed a limitation. For more than half of the participants, we know the age at which they stopped. For the rest of the enrolled participants, we only as-sumed they were adults based on appearance and previous upload history.

We have added this limitation to the strengths and limitations section.: “Furthermore, for a large proportion of our study participants, we had to assume from looks and past upload history that they were adults.” (lines 464-465)

p.11 line 200 where did you obtain ethical clearance to conduct this study? I under-stand that informed consent wasn't a requirement (as per lines 202 - 203). Also, please expand on the ethical appropriateness of conducting the study without informed consent from the Youtubers.

Authors: We have added our ethical approval for the study. In addition, we have further considered the ethical appropriateness of our data: “The ethical approval for this study has been granted by the Ethical Review Committee of the Medical Faculty at Martin Luther University Halle-Wittenberg.” (lines 211- 212).

And have extended a little bit on the ethical appropriateness of conducting the study: “Before a person uploads a video on YouTube, they have to c

---

## [Editor Report · Decision Letter 2]

2 Apr 2024

YouTube and the implementation and discontinuation of the oral contraceptive pill: A mixed-method content analysis

PONE-D-24-01385R2

Dear Dr. Nieman

We’re pleased to inform you that your manuscript has been judged scientifically suitable for publication and will be formally accepted for publication once it meets all outstanding technical requirements.

Kind regards,

Deidre Pretorius, PhD

Academic Editor

PLOS ONE

Additional Editor Comments (optional):

Interesting research!
---

## [Editor Report · Acceptance letter]

15 May 2024

PONE-D-24-01385R2 

PLOS ONE

Dear Dr. Niemann, 

I'm pleased to inform you that your manuscript has been deemed suitable for publication in PLOS ONE. Congratulations! Your manuscript is now being handed over to our production team.

Kind regards, 

on behalf of

Dr. Deidre Pretorius 

Academic Editor

PLOS ONE